# Optimization Production of an Endo-β-1,4-Xylanase from *Streptomyces thermocarboxydus* Using Wheat Bran as Sole Carbon Source

Thi Ngoc Tran [1], Chien Thang Doan [1], Thi Kieu Loan Dinh [2], Thi Hai Ninh Duong [1], Thi Thuc Uyen Phan [1], Thi Thuy Loan Le [1], Trung Dung Tran [1], Pham Hung Quang Hoang [1], Anh Dzung Nguyen [3] and San-Lang Wang [4,5,*]

1 Faculty of Natural Science and Technology, Tay Nguyen University, Buon Ma Thuot 630000, Vietnam; ttngoc@ttn.edu.vn (T.N.T.); dcthang@ttn.edu.vn (C.T.D.); dthninh@ttn.edu.vn (T.H.N.D.); pttuyen@ttn.edu.vn (T.T.U.P.); lttloan@ttn.edu.vn (T.T.L.L.); ttdung@ttn.edu.vn (T.D.T.); hphquang@ttn.edu.vn (P.H.Q.H.)
2 Faculty of Education, Tay Nguyen University, Buon Ma Thuot 630000, Vietnam; dtkloan@ttn.edu.vn
3 Institute of Biotechnology and Environment, Tay Nguyen University, Buon Ma Thuot 630000, Vietnam; nadzung@ttn.edu.vn
4 Department of Chemistry, Tamkang University, New Taipei City 25137, Taiwan
5 Life Science Development Center, Tamkang University, New Taipei City 25137, Taiwan
* Correspondence: sabulo@mail.tku.edu.tw

**Abstract:** Xylanases, key enzymes for hydrolyzing xylan, have diverse industrial applications. The bioprocessing of agricultural byproducts to produce xylanase through fermentation approaches is gaining importance due to its significant potential to reduce enzyme production costs. In this work, the productivity of *Streptomyces thermocarboxydus* TKU045 xylanase was enhanced through liquid fermentation employing wheat bran as the sole carbon source. The maximum xylanase activity ($25.314 \pm 1.635$ U/mL) was obtained using the following optima factors: 2% ($w/v$) wheat bran, 1.4% ($w/v$) $KNO_3$, an initial pH of 9.8, an incubation temperature of 37.3 °C, and an incubation time of 2.2 days. Xylanase (Xyn_TKU045) of 43 kDa molecular weight was isolated from the culture supernatant and was biochemically characterized. Analysis through liquid chromatography with tandem mass spectrometry revealed a maximum amino acid identity of 19% with an endo-1,4-β-xylanase produced by *Streptomyces lividans*. Xyn_TKU045 exhibited optimal activity at pH 6, with remarkable stability within the pH range of 6.0 to 8.0. The enzyme demonstrated maximum efficiency at 60 °C and considerable stability at ≤70 °C. $Mg^{2+}$, $Mn^{2+}$, $Ba^{2+}$, $Ca^{2+}$, 2-mercaptoethanol, Tween 20, Tween 40, and Triton X-100 positively influenced Xyn_TKU045, while $Zn^{2+}$, $Fe^{2+}$, $Fe^{3+}$, $Cu^{2+}$, and sodium dodecyl sulfate exhibited adverse impact. The kinetic properties of Xyn_TKU045 were a $K_m$ of 0.628 mg/mL, a $k_{cat}$ of 75.075 $s^{-1}$ and a $k_{cat}/K_m$ of 119.617 mL $mg^{-1}s^{-1}$. Finally, Xyn_TKU045 could effectively catalyze birchwood xylan into xylotriose and xylobiose as the major products.

**Keywords:** agricultural residue; optimization production; *Streptomyces thermocarboxydus*; wheat bran; xylanase

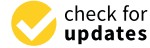



## 1. Introduction

Xylanases, classified as endo-1,4-β-d-xylanohydrolase (EC 3.2.1.8), are hemicellulases commonly employed in processes such as pulp bleaching and paper deinking [1]. Furthermore, these enzymes exhibit remarkable versatility and can be used in diverse sectors such as alternative fuel generation, baking, prebiotic preparation, and the clarification of fruit juices [2–6]. The source of this enzyme has primarily been associated with various fungi (for example, *Fusarium* [2], *Trichoderma* [7], *Aspergillus* [8], *Thielavia* [9], and *Rhizomucor* [10]), and bacteria (for example, *Bacillus* [4], *Streptomyces* [11], *Paenibacillus* [12], and *Halomonas* [13]), showcasing the diverse microbial sources of this enzyme. Among

the strains, *Streptomyces*, a bacterial genus renowned for its prolific production of various enzymes, including xylanases, holds promise for industrial applications. The unique characteristics of *Streptomyces* make it an interesting subjects for further research and development in the section of enzymology and biotechnology. This species was reported for its capacity to generate thermostable xylanases that remain active at elevated temperatures ($\geq$60 °C) [14,15]. *Streptomyces* strains also exhibit the capability to utilize a diverse array of carbon and nitrogen sources, including various types of residues, for the biosynthesis of enzymes [16–20], including xylanase [21–23].

The utilization of xylan as the source of carbon and an inducer for xylanase production poses certain economic challenges due to its high cost of production. To address this issue, researchers have explored substituting commercially available xylan with low-cost and readily available agricultural by-products, resulting in a significant reduction in the cost of overall processing [24]. This innovative approach, utilizing abundant agricultural residues, opens avenues for the cost-effective production of xylanase. By tapping into these agricultural residues, the production costs can be minimized and the sustainable utilization of waste materials can be promoted, aligning with eco-friendly practices in enzyme production. A vast array of agricultural residues, such as sugarcane bagasse [25], rice straw [26], rice bran [27], wheat bran [28], coconut husk [29], corn stalk [30], and corncob [31], can be used as the carbon source or both carbon and nitrogen sources for microbial xylanase synthesis.

Maximizing productivity yield is an essential step for the industrial applications of enzymes. Culture conditions can be optimized using the one-factor-at-a-time (OFAT) method. However, this approach does not account for interactions between factors. To address this limitation, response surface methodology (RSM) has been effectively used as an alternative in optimization solutions. Indeed, RSM has been effectively applied to maximize xylanase productivity [26]. In a previous study, a potential xylanase-producing strain, *Streptomyces thermocarboxydus* TKU045 demonstrated the ability to utilize wheat bran powder (WBP) as the exclusive carbon source and exhibited elevated xylanase activity [32]. However, the enzyme production was not statistically optimized, and the purification process was not performed at that time. Therefore, our objective was to figure out the cultivation conditions for maximizing xylanase productivity using RSM combined with Box–Behnken design (BBD), followed by its purification for subsequent biochemical characterization.

## 2. Results and Discussion

*2.1. Production Optimization of Streptomyces thermocarboxydus TKU045 Xylanase Using Wheat Bran Powder as the Sole Carbon Source*

Optimizing medium and culture parameters is crucial for a significant improvement in xylanase productivity, a goal achievable through response surface methodology (RSM) [3,4,33]. This approach comprehensively examines the interactive impacts of all independent factors in a fermentation process and investigates the specific interactions between a response variable and a series of design-independent variables [3], which the OFAT method cannot achieve. The Box–Behnken design (BBD), a top-rated design of RSM [34], was employed to optimize the xylanase productivity of *S. thermocarboxydus* TKU045, incorporating five factors: the amount of WBP ($X_1$), the amount of KNO$_3$ ($X_2$), initial pH ($X_3$), incubation temperature ($X_4$), and incubation time ($X_5$). The BBD comprises of 46 runs; the highest xylanolytic activity (26.978 U/mL) was noted at run 42 (Table 1). The data were subjected to regression fitting to derive the regression formula for the second-order model:

$$Y \text{ (xylanase activity, U/mL)} = -2167.2 + 39.162X_1 + 138.53X_2 + 180.64X_3 + 61.012X_4 + 34.787X_5 - 11.173X_1 \times X_1 - 27.745X_2 \times X_2 - 9.016X_3 \times X_3 - 0.781X_4 \times X_4 - 10.187X_5 \times X_5 - 0.375X_1 \times X_2 + 0.431X_1 \times X_3 - 0.042X_1 \times X_4 + 1.680X_1 \times X_5 - 1.115X_2 \times X_3 - 1.261X_2 \times X_4 - 1.678X_2 \times X_5 - 0.131X_3 \times X_4 + 0.548X_3 \times X_5 + 0.122X_4 \times X_5$$

**Table 1.** The results of the Box–Behnken design experiment.

| Run | Coded Levels | | | | | Uncoded Levels | | | | | Y (U/mL) |
|---|---|---|---|---|---|---|---|---|---|---|---|
| | $X_1$ | $X_2$ | $X_3$ | $X_4$ | $X_5$ | $X_1$ | $X_2$ | $X_3$ | $X_4$ | $X_5$ | |
| 1 | −1 | −1 | 0 | 0 | 0 | 1 | 1 | 10 | 37 | 2 | 8.494 |
| 2 | +1 | −1 | 0 | 0 | 0 | 3 | 1 | 10 | 37 | 2 | 9.396 |
| 3 | −1 | +1 | 0 | 0 | 0 | 1 | 2 | 10 | 37 | 2 | 1.500 |
| 4 | +1 | +1 | 0 | 0 | 0 | 3 | 2 | 10 | 37 | 2 | 1.651 |
| 5 | 0 | 0 | −1 | −1 | 0 | 2 | 1.5 | 9 | 34 | 2 | 14.488 |
| 6 | 0 | 0 | +1 | −1 | 0 | 2 | 1.5 | 11 | 34 | 2 | 0.043 |
| 7 | 0 | 0 | −1 | +1 | 0 | 2 | 1.5 | 9 | 40 | 2 | 16.373 |
| 8 | 0 | 0 | +1 | +1 | 0 | 2 | 1.5 | 11 | 40 | 2 | 0.355 |
| 9 | 0 | −1 | 0 | 0 | −1 | 2 | 1 | 10 | 37 | 1 | 2.613 |
| 10 | 0 | +1 | 0 | 0 | −1 | 2 | 2 | 10 | 37 | 1 | 0.561 |
| 11 | 0 | −1 | 0 | 0 | +1 | 2 | 1 | 10 | 37 | 3 | 14.571 |
| 12 | 0 | +1 | 0 | 0 | +1 | 2 | 2 | 10 | 37 | 3 | 9.163 |
| 13 | −1 | 0 | −1 | 0 | 0 | 1 | 1.5 | 9 | 37 | 2 | 6.858 |
| 14 | +1 | 0 | −1 | 0 | 0 | 3 | 1.5 | 9 | 37 | 2 | 5.093 |
| 15 | −1 | 0 | +1 | 0 | 0 | 1 | 1.5 | 11 | 37 | 2 | 0.086 |
| 16 | +1 | 0 | +1 | 0 | 0 | 3 | 1.5 | 11 | 37 | 2 | 0.045 |
| 17 | 0 | 0 | 0 | −1 | −1 | 2 | 1.5 | 10 | 34 | 1 | 0.365 |
| 18 | 0 | 0 | 0 | +1 | −1 | 2 | 1.5 | 10 | 40 | 1 | 0.624 |
| 19 | 0 | 0 | 0 | −1 | +1 | 2 | 1.5 | 10 | 34 | 3 | 10.960 |
| 20 | 0 | 0 | 0 | +1 | +1 | 2 | 1.5 | 10 | 40 | 3 | 12.687 |
| 21 | 0 | −1 | −1 | 0 | 0 | 2 | 1 | 9 | 37 | 2 | 18.841 |
| 22 | 0 | +1 | −1 | 0 | 0 | 2 | 2 | 9 | 37 | 2 | 12.625 |
| 23 | 0 | −1 | +1 | 0 | 0 | 2 | 1 | 11 | 37 | 2 | 8.638 |
| 24 | 0 | +1 | +1 | 0 | 0 | 2 | 2 | 11 | 37 | 2 | 0.193 |
| 25 | −1 | 0 | 0 | −1 | 0 | 1 | 1.5 | 10 | 34 | 2 | 8.998 |
| 26 | +1 | 0 | 0 | −1 | 0 | 3 | 1.5 | 10 | 34 | 2 | 6.257 |
| 27 | −1 | 0 | 0 | +1 | 0 | 1 | 1.5 | 10 | 40 | 2 | 9.622 |
| 28 | +1 | 0 | 0 | +1 | 0 | 3 | 1.5 | 10 | 40 | 2 | 6.374 |
| 29 | 0 | 0 | −1 | 0 | −1 | 2 | 1.5 | 9 | 37 | 1 | 2.645 |
| 30 | 0 | 0 | +1 | 0 | −1 | 2 | 1.5 | 11 | 37 | 1 | 0.000 |
| 31 | 0 | 0 | −1 | 0 | +1 | 2 | 1.5 | 9 | 37 | 3 | 11.229 |
| 32 | 0 | 0 | +1 | 0 | +1 | 2 | 1.5 | 11 | 37 | 3 | 10.778 |
| 33 | −1 | 0 | 0 | 0 | −1 | 1 | 1.5 | 10 | 37 | 1 | 0.526 |
| 34 | +1 | 0 | 0 | 0 | −1 | 3 | 1.5 | 10 | 37 | 1 | 0.680 |
| 35 | −1 | 0 | 0 | 0 | +1 | 1 | 1.5 | 10 | 37 | 3 | 4.983 |
| 36 | +1 | 0 | 0 | 0 | +1 | 3 | 1.5 | 10 | 37 | 3 | 11.855 |
| 37 | 0 | −1 | 0 | −1 | 0 | 2 | 1 | 10 | 34 | 2 | 12.385 |
| 38 | 0 | +1 | 0 | −1 | 0 | 2 | 2 | 10 | 34 | 2 | 8.163 |
| 39 | 0 | −1 | 0 | +1 | 0 | 2 | 1 | 10 | 40 | 2 | 18.121 |
| 40 | 0 | +1 | 0 | +1 | 0 | 2 | 2 | 10 | 40 | 2 | 6.335 |
| 41 | 0 | 0 | 0 | 0 | 0 | 2 | 1.5 | 10 | 37 | 2 | 26.148 |
| 42 | 0 | 0 | 0 | 0 | 0 | 2 | 1.5 | 10 | 37 | 2 | 26.978 |
| 43 | 0 | 0 | 0 | 0 | 0 | 2 | 1.5 | 10 | 37 | 2 | 22.187 |
| 44 | 0 | 0 | 0 | 0 | 0 | 2 | 1.5 | 10 | 37 | 2 | 22.402 |
| 45 | 0 | 0 | 0 | 0 | 0 | 2 | 1.5 | 10 | 37 | 2 | 26.188 |
| 46 | 0 | 0 | 0 | 0 | 0 | 2 | 1.5 | 10 | 37 | 2 | 23.790 |

$X_1$, amount of wheat bran powder (WBP; %); $X_2$, amount of $KNO_3$ (%); $X_3$, initial pH of medium; $X_4$, incubation temperature (°C); $X_5$, incubation time (day).

This regression equation was obtained using the result (mentioned in Table 2) generated by analyzing experimental data with the RSM function [35]. According to Table 2, the low *p*-value ($p < 0.0001$) and significant F-value (15.15) indicate that the model is significant. The high $R^2$ (0.9238) suggests that the model fits well with the data; only 7.62% of total variations could not be described by the model. In addition, the adjusted $R^2$ (0.8628) was also high, confirming the model's significance. The analysis of variance based upon the *p*-value suggested that the linear terms ($X_2$, $X_3$, and $X_4$) and the quadratic terms ($X_1^2$,

$X_2^2$, $X_3^2$, $X_4^2$, and $X_5^2$) were significant. The lack of fit for the proposed model was not significant, indicating that it is fit for prediction, and hence the statistical implication of this theoretical account was turned over by the equations for coded factors (Table 3). As the two-factor interaction term was insignificant, the model was refined by removing this term from the regression formula to obtain the refined model, as follows:

$$Y = -2058.124 + 44.708X_1 + 76.627X_2 + 176.077X_3 + 57.970X_4 + 45.638X_4$$
$$- 11.173X_1^2 - 27.745X_2^2 - 9.016X_3^2 - 0.781X_4^2 - 10.187X_5^2$$

with an $R^2 = 0.9125$, an adjusted $R^2 = 0.8875$, an F-value = 36.5, and a *p*-value < 0.00001.

**Table 2.** Results of regression analysis using the Box–Behnken design.

| Term | Coefficient Estimate | Standard Error Coefficient | t Value | Pr (>|t|) | |
|---|---|---|---|---|---|
| Constant | −2167.200 | 296.800 | −7.302 | <0.00001 | *** |
| $X_1$ | 39.162 | 24.626 | 1.590 | 0.124 | |
| $X_2$ | 138.530 | 49.627 | 2.791 | 0.010 | ** |
| $X_3$ | 180.640 | 28.006 | 6.450 | <0.00001 | *** |
| $X_4$ | 61.012 | 9.894 | 6.167 | <0.00001 | *** |
| $X_5$ | 34.787 | 24.626 | 1.413 | 0.170 | |
| $X_1X_2$ | −0.375 | 2.983 | −0.126 | 0.901 | |
| $X_1X_3$ | 0.431 | 1.491 | 0.289 | 0.775 | |
| $X_1X_4$ | −0.042 | 0.497 | −0.085 | 0.933 | |
| $X_1X_5$ | 1.680 | 1.491 | 1.126 | 0.271 | |
| $X_2X_3$ | −1.115 | 2.983 | −0.374 | 0.712 | |
| $X_2X_4$ | −1.261 | 0.994 | −1.268 | 0.216 | |
| $X_2X_5$ | −1.678 | 2.983 | −0.563 | 0.579 | |
| $X_3X_4$ | −0.131 | 0.497 | −0.264 | 0.794 | |
| $X_3X_5$ | 0.548 | 1.491 | 0.368 | 0.716 | |
| $X_4X_5$ | 0.122 | 0.497 | 0.246 | 0.808 | |
| $X_1^2$ | −11.173 | 1.010 | −11.066 | <0.00001 | *** |
| $X_2^2$ | −27.745 | 4.038 | −6.870 | <0.00001 | *** |
| $X_3^2$ | −9.016 | 1.010 | −8.930 | <0.00001 | *** |
| $X_4^2$ | −0.781 | 0.112 | −6.961 | <0.00001 | *** |
| $X_5^2$ | −10.187 | 1.010 | −10.091 | <0.00001 | *** |

$R^2 = 0.9238$; Adjusted $R^2 = 0.8628$; F-statistic: 15.15 on 20 and 25 degrees of freedom (DF); *p*-value: $2.11 \times 10^{-9}$; Significance codes: 0 '***' 0.001 '**' 0.01. $X_1$, amount of wheat bran powder (WBP; %); $X_2$, amount of $KNO_3$ (%); $X_3$, initial pH of medium; $X_4$, incubation temperature (°C); $X_5$, incubation time (day).

**Table 3.** The results of the analysis of variance for the fitted quadratic polynomial model.

| Source | Degrees of Freedom | Sum of Squares | Mean Square | F Value | Pr (>F) |
|---|---|---|---|---|---|
| Fo | 5 | 851.01 | 170.20 | 19.133 | <0.00001 |
| TWI | 10 | 32.95 | 3.30 | 0.371 | 0.948 |
| PQ | 5 | 1811.54 | 362.31 | 40.729 | <0.00001 |
| Residuals | 25 | 222.39 | 8.90 | | |
| Lack of fit | 20 | 200.51 | 10.03 | 2.291 | 0.182 |
| Pure error | 5 | 21.88 | 4.38 | | |

The interplay between two factors, with the others maintained at their midpoint (0) level, is represented in the two-dimensional (2D) contour plots (Figure S1) and 3D response surface plots (Figure S2). These plots exhibit peak xylanase activity near the midpoints (with 2% WBP, 1.5% $KNO_3$, pH 10, a temperature of 37 °C, and an incubation time of 2 days), indicating that the designated levels were within the suitable range. There was an interesting alignment between the plots and regression analysis, suggesting that the interaction among the factors was insignificant. Noticeably, increasing the values of the

factors led to a rise in xylanase productivity until the maximum value was achieved. However, a continued increase in these factors resulted in a corresponding decline in xylanase productivity. The optimal levels of WBP, pH, incubation temperature, and incubation time were estimated as 2%, 1.4%, 9.8, 37.3 °C, and 2.2 days, respectively, with a predicted xylanase productivity of 26.137 U/mL. Accordingly, a confirmation experiment was carried out by cultivating *S. thermocarboxydus* TKU045 under optimal conditions. The actual value of xylanase productivity was 25.314 ± 1.635 U/mL, which did not significantly differ from the predicted value (26.137 U/mL). There were relatively few reports on the statistical optimization of xylanase production from *Streptomyces* genus. By using the Central composite design of RSM, the optimal conditions for the xylanase production of *Streptomyces* sp. ER1 were 0.37% xylan and 33.10 mg/L olive oil [3]. The ideal conditions for the xylanase production of *Streptomyces variabilis* (MAB3), using the Box–Behnken design, were found to be 2% birchwood xylan, pH 8.2, a temperature of 46.5 °C, and an incubation time of 68 h [36]. Recently, Medouni-Haroune et al. (2024) reported the optimal conditions for the xylanase production of *Streptomyces* sp. S1M3I using the Box–Behnken design of RSM, which were 3% olive pomace powder, 0.3% $(NH_4)_2SO_4$, pH 7.4, and an incubation temperature of 40 °C [37]. It is evident that the optimal conditions for xylanase production vary. Therefore, determining the optimal enzyme production conditions for each strain of *Streptomyces* is an essential step. Furthermore, the optimization of xylanase produced by *S. thermocarboxydus* is rarely reported. Taken together, this study could be a novel observation of xylanase production optimization from the species *Streptomyces thermocarboxydus*.

### 2.2. Enzyme Purification

*S. thermocarboxydus* TKU045 was grown on a WBP-containing medium under optimized conditions. Subsequently, 1 L of the culture supernatant was harvested and precipitated with cold ethanol (−20 °C). While other studies commonly employ $(NH_4)_2SO_4$ to concentrate xylanase, our study reveals that xylanase activity through $(NH_4)_2SO_4$ precipitation was significantly low. In contrast, cold ethanol could retain nearly all xylanase activity (Table 4). Therefore, cold ethanol was employed as a substitute for $(NH_4)_2SO_4$. The crude enzyme was then loaded onto a High-Q column for initial purification. The chromatography profile for the separation of three xylanase activity peaks (F1, F2, and F3) with a NaCl gradient in the range of 0 to 0.5 M is shown in Figure 1a. This result aligned with our previous findings, in which the cultivation medium of *S. thermocarboxydus* TKU045 displayed multiple bands of xylanolytic activity on PAGE gel containing xylan [32]. All three peaks were desalted by dialysis and further purified using a DEAE sepharose column. However, only fraction F1 could be successfully purified, yielding a single protein band, as shown by SDS-PAGE (Figure 1b). In contrast, fractions F2 and F3 lost their activity completely during the subsequent purification step. Consequently, only one xylanase (Xyn_TKU045) was successfully purified.

**Table 4.** A summary of the purification of Xyn-TKU045.

| Step | Total Protein (mg) | Total Activity (U) | Specific Activity (U/mg) | Recovery (%) | Purification (Fold) |
|---|---|---|---|---|---|
| Cultural supernatant | 4858.7 | 26,111.0 | 5.4 | 100 | 1.0 |
| Ethanol precipitation | 2439.1 | 25,078.6 | 10.3 | 96 | 1.9 |
| High Q column | 57.7 | 10,857.8 | 188.1 | 42 | 35.0 |
| DEAE sepharose column | 3.2 | 3882.330 | 1212.7 | 15 | 225.7 |

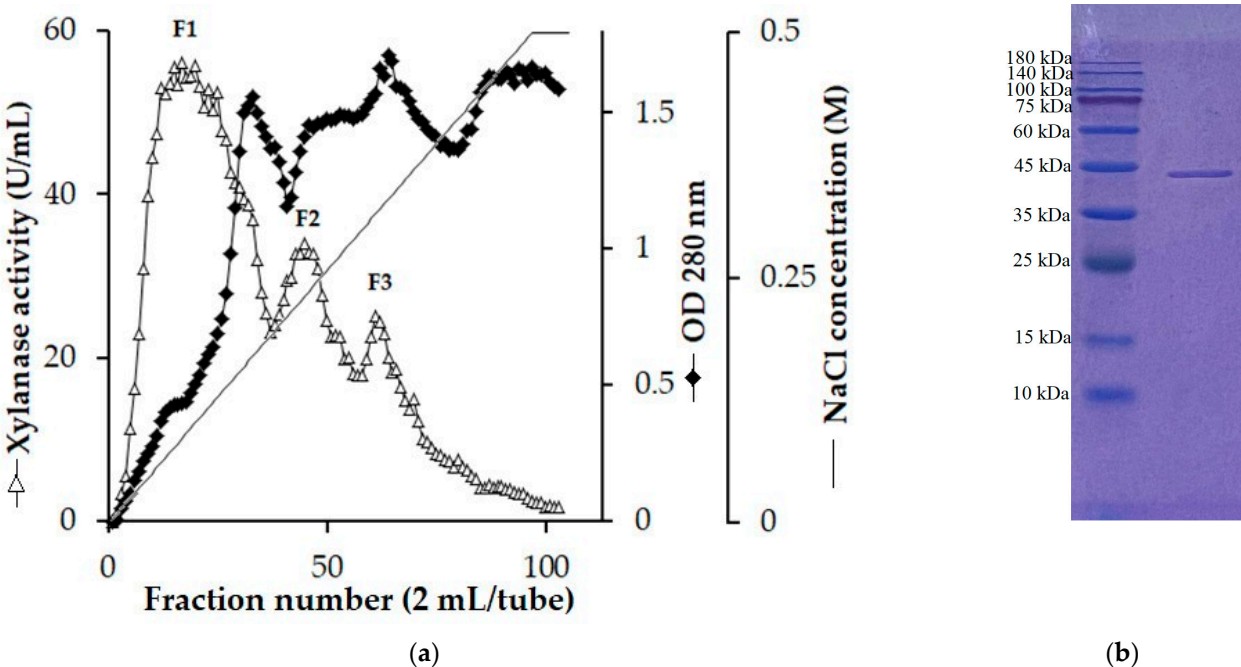

**Figure 1.** Chromatography profile of the crude enzyme on High-Q column (**a**), and sodium dodecyl sulfate-polyacrylamide gel electrophoresis profile of Xyn-TKU045 (**b**).

The molecular weight (MW) of Xyn_TKU045 was approximately 43 kDa (Figure 1b). This falls within the typical range of 15 kDa to 50 kDa for xylanases found in *Streptomyces* [32]. The MW of Xyn_TKU045 was comparable to that of *S. thermocarboxydus* HY-5's xylanase (43.962 kDa) [38] and markedly inferior to that of *S. thermocarboxydus* MW8's xylanase (52 kDa) [39].

To further verify the identity of the purified enzyme, the band corresponding to Xyn_TKU045 was excised from the SDS-PAGE gel, digested using trypsin, and subsequently analyzed using the LC-MS/MS method. The MASCOT search results, utilizing the Swissprot database and Firmicutes taxonomy, indicated the close association of xylanase with endo-1,4-beta-xylanase. The closest relative was identified as XYNA_STRLI (*Streptomyces lividans*) with 19% amino acid sequence identity (Table 5).

**Table 5.** Identification of Xyn_TKU045 using liquid chromatography with tandem mass spectrometry analysis.

| Matched Peptide Sequence | Identified Protein and Coverage Rate | Strain |
|---|---|---|
| $^{90}$IDATEPQR$^{97}$ $^{144}$QAMIDHINGVMAHYK$^{158}$ $^{161}$IVQWDVVNEA FADGSSGAR$^{179}$ $^{208}$LCYNDYNVENWTWAK$^{222}$ $^{247}$GVPIDCVGFQSHFNSGSPYNSNFR$^{260}$ $^{418}$VQIYSCWGGDNQK$^{430}$ | Endo-1,4-beta-xylanase 19% | *Streptomyces lividans* |

*2.3. Biochemical Characterization*

As illustrated in Figure 2a, the optimal pH of Xyn_TKU045 was determined to be pH 6 when assessed in a phosphate buffer. The enzyme exhibited remarkable activity, with over 80% maintained at both pH 7 and 8. This result suggests the versatility and effectiveness of the Xyn_TKU045 across a broad optimal pH spectrum from pH 6 to pH 8. Moreover, Xyn_TKU045 demonstrated robust stability within the pH range of 6 to 8. Studies published earlier have consistently found that the optimal pH for xylanases from

*Streptomyces* is typically from pH 5 to pH 7 [40–42]. The broad optimal pH range and stability observed in Xyn_TKU045 highlight its potential to function effectively under diverse pH conditions.

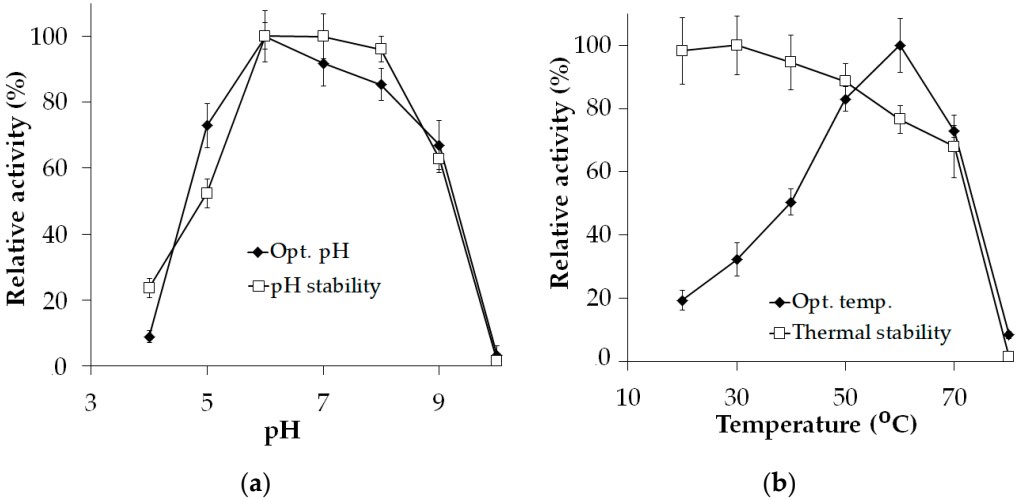

(a)          (b)

**Figure 2.** Effect of pH (**a**), and temperature (**b**) on the activity of xylanase Xyn_TKU045.

As illustrated in Figure 2b, the optimal temperature of Xyn_TKU045 was noted at 60 °C. When incubated at various temperature levels, Xyn_TKU045 maintained over 75% activity at temperatures up to 60 °C, while at 70 °C, the enzyme still retained approximately 68%. Thus, according to this study, Xyn_TKU045 is categorized as a thermophilic enzyme. Additionally, this xylanase displayed thermal stability (up to 60 °C) and activity (60 °C) comparable to or even better than most *Streptomyces* xylanases [21,40,43,44].

The impact of metal ions on Xyn_TKU045 was studied and the results are mentioned in Table 6. Metal ions that exhibited inhibitory effects include $Zn^{2+}$ (2.676%), $Fe^{2+}$ (51.140%), $Fe^{3+}$ (0.793%), and $Cu^{2+}$ (0.000%), whereas those that activate the enzyme comprise $Mg^{2+}$ (142.616%), $Mn^{2+}$ (219.029%), $Ba^{2+}$ (168.682%), and $Ca^{2+}$ (170.565%). Metal ions may exhibit variable effects among xylanases originating from different microbial strains. For instance, $Ca^{2+}$ and $Mn^{2+}$ exerted inhibitory effects on *Streptomyces matensis* DW67's xylanase [42], but they improved the activity of Xyn_TKU045 (this study) and *Streptomyces thermovulgaris* TISTR1948 [40]. Xyn_TKU045 was inactivated by Sodium lauryl sulfate (SDS), an anionic detergent with robust protein denaturing properties, highlighting the significance of hydrophobic interactions in preserving the 3D conformation of the protein. On the other hand, nonionic detergents (Tween 20, Tween 40, and Triton X-100) stimulated xylanase activity (164.420%, 161.943%, and 149.653%, respectively). The nonionic detergents enhance the disaggregation of proteins, thereby improving the hydrolysis activity of enzymes through the exposure of their catalytic sites. The exploration of the effect of 2-mercaptoethanol, a reducing agent, on enzyme activity revealed that it could simulate the activity of Xyn_TKU045 (166.501%) (Table 6). While 2-mercaptoethanol has already been recognized as a xylanase activity enhancer for certain *Streptomyces* strains [42,44], the xylanase derived from Xyn_TKU045 was unaffected by (Ethylenedinitrilo)tetraacetic acid (EDTA) (with a relative activity of 105.198 ± 4.454%), indicating that it might not be a metalloenzyme.

Xyn_TKU045 demonstrated notable xylanolytic activity on three kinds of xylans (Table 7). However, Xyn_TKU045 could not hydrolyze non-xylan substrates such as starch, cellulose, and pectin. This substrate specificity emphasizes the tailored functionality of the enzyme, making it a promising candidate for applications that specifically target xylan-containing materials. Different amounts of birchwood xylan were used to determine the kinetic parameters for Xyn_TKU045. Accordingly, the $K_m$ (reflects substrate affinity), $k_{cat}$ (reflects catalytic rate), and $k_{cat}/K_m$ (reflects catalytic efficiency) values for Xyn_TKU045 were 0.628 mg/mL, 75.075 $s^{-1}$, and 119.617 mL $mg^{-1}s^{-1}$, respectively. Li et al. (2022)

investigated the kinetic properties of xylanase derived from *Streptomyces* sp. T7 and found that the $K_m$ and $k_{cat}/K_m$ values for its activity on birchwood xylan were 2.78 mg/mL and 42.91 $s^{-1}mg^{-1}$, respectively [14]. In another report, the $K_m$, $k_{cat}$, and $k_{cat}/K_m$ values for XynA from *S. rameus* L2001 were found to be 19.18 mg/mL, 1208.00 $s^{-1}$, 62.98 mL $mg^{-1}s^{-1}$, respectively [45].

**Table 6.** Effect of various chemicals on the activity of xylanase Xyn_TKU045.

| Chemical | Relative Activity (%) |
|---|---|
| Control | $100.000 \pm 10.057$ |
| $Zn^{2+}$ | $2.676 \pm 4.837$ |
| $Fe^{2+}$ | $51.140 \pm 9.080$ |
| $Fe^{3+}$ | $0.793 \pm 6.868$ |
| $Cu^{2+}$ | $0.000 \pm 3.156$ |
| $Mg^{2+}$ | $142.616 \pm 10.016$ |
| $Mn^{2+}$ | $219.029 \pm 18.431$ |
| $Ba^{2+}$ | $168.682 \pm 18.554$ |
| $Ca^{2+}$ | $170.565 \pm 11.894$ |
| 2-mercaptoethanol | $166.501 \pm 8.121$ |
| Tween 20 | $164.420 \pm 5.252$ |
| Tween 40 | $161.943 \pm 27.468$ |
| Triton X-100 | $149.653 \pm 13.872$ |
| SDS | $3.964 \pm 5.152$ |
| EDTA | $105.198 \pm 4.454$ |

**Table 7.** Substrate specificity of xylanase Xyn_TKU045.

| Chemical | Relative Activity (%) |
|---|---|
| Birchwood xylan * | $100.000 \pm 6.585$ |
| Beechwood xylan | $93.433 \pm 6.935$ |
| Oatspelt xylan | $95.646 \pm 5.945$ |
| Starch | ND |
| Pectin | ND |
| Cellulose | ND |

*, served as the control; ND, no xylanase activity detected.

### 2.4. Hydrolysis Pattern and Xylooligosaccharide Production

The xylan hydrolysis products after Xyn_TKU045 catalysis were analyzed using HPLC. As shown in Figure 3, after 30 min of hydrolysis, the obtained products were a mixture of xylooligosaccharides (XOSs). Upon extending the hydrolysis time, peaks corresponding to xylobiose ($X_2$) and xylobiose ($X_3$) were formed and became the predominant products. This outcome indicates that Xyn_TKU045 is an endo-enzyme. Additionally, starting from 2 h onwards, the HPLC analysis revealed a peak at the xylose position, suggesting that Xyn_TKU045 also exhibits exo-enzyme activity, albeit to a lesser extent. The presence of this xylose peak after 2 h implies that the exo-enzyme activity gradually manifested, providing insights into the enzyme's versatility in catalyzing both endo- and exo-type reactions during xylan hydrolysis. Numerous xylanases from *Streptomyces* have also been reported to primarily generate $X_2$ and $X_3$ during xylan hydrolysis [21,46–49].

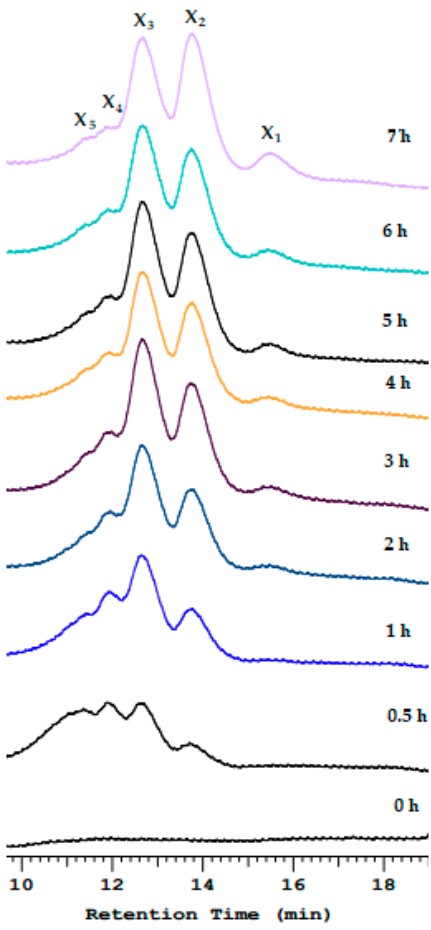

**Figure 3.** Hydrolysis pattern of *Streptomyces thermocarboxydus* TKU045 xylanase toward birchwood xylan. $X_1$, xylose; $X_2$, xylobiose; $X_3$, xylotriose; $X_4$, xylotetraose; $X_5$, xylopentose.

Low-molecular-weight XOSs, including $X_2$ and $X_3$, present significant commercial value as emerging prebiotics [50]. In a previous study, XOSs derived from xylan hydrolysis using *S. thermocarboxydus* TKU045's crude enzyme cocktail exhibited antioxidant and prebiotic activities on the *Bifidobacterium bifidum* BCRC 14615 [32]. Thus, it is interesting to explore the potential of Xyn_TKU045 as the biocatalyst for XOS production. To prepare XOSs, birchwood xylan (1%) was hydrolyzed by Xyn_TKU045 at different incubation times (from 0 h to 24 h). As illustrated in Figure S3, the concentration of reducing sugars reached its peak at 7.316 mg/mL after 6 h of incubation. Following this incubation period, the hydrolysate was subjected to HPLC analysis. The results indicated that 96% of hydrolysis products consisted of XOSs, while xylose accounted for only 4% of the total content. Interestingly, the analysis revealed that the majority of the XOSs were oligomers, with dimers and trimers making up 59% of the total XOSs present. Overall, the findings highlight that xylanase from *S. thermocarboxydus* TKU045 holds promise for producing bioactive low-molecular-weight XOSs.

## 3. Materials and Methods

### 3.1. Materials

The strain *Streptomyces thermocarboxydus* TKU045 was the same as was used in our previous work [32]. Xylose, xylan, and 3,5-dinitrosalicylic acid were obtained from Sigma Co. (St. Louis, MO, USA). Wheat bran was obtained from Miaoli (Miaoli City, Taiwan). Other chemicals were of the highest grade of purity.

### 3.2. Xylanase Assay and Protein Determination

The xylanase assay was conducted according to the method of Miller (1959) [51] using birchwood xylan [32]. Overall, the reaction component comprised 200 μL birchwood xylan (1%, pH 6, prepared sodium phosphate buffer) and 50 μL enzyme solution. This blend underwent incubation at 70 °C for 60 min, and 1500 μL DNS reagent was added and its concentration of reducing sugar was estimated through optical density (515 nm) measurement. An xylanase unit is determined by the enzyme quantity necessary to catalyze the liberation of 1 μmol product (equivalent to xylose) within one minute [32]. The protein was ascertained using the method of Lowry et al. (1951) [52].

### 3.3. Optimization of Production

The Box–Behnken design was employed for optimizing the response associated with five independent factors, namely, $X_1$ (the amount of WBP, % $w/v$), $X_2$ (the amount of $KNO_3$, % $w/v$), $X_3$ (the initial pH of the medium), $X_4$ (incubation temperature, °C), and $X_5$ (incubation time, days). The process was optimized using R-software (version 2021.09.1+372), with the "rsm" package employed for data analysis as well as graphical representation [35]. Forty-six runs, including 6 center points, were executed for optimization, and each factor was tested at three levels, low, medium, and high, represented, respectively, by the coded values of −1, 0, and +1. The determination of factor levels was guided by preliminary results obtained from experiments involving one-factor-at-a-time (OFAT) [32]. The resulting data were analyzed using R-software to establish the optimal conditions for xylanase production. The impact of the five examined factors was symbolized by the following quadratic formula:

$$
\begin{aligned}
Y = {}& \beta_0 + \beta_1 X_1 + \beta_2 X_2 + \beta_3 X_3 + \beta_4 X_4 + \beta_5 X_5 + \beta_{12} X_1 \times X_2 + \beta_{13} X_1 \times X_3 + \beta_{14} X_1 \\
& \times X_4 + \beta_{15} X_1 \times X_5 + \beta_{23} X_2 \times X_3 + \beta_{24} X_2 \times X_4 + \beta_{25} X_2 \times X_5 + \beta_{34} X_3 \times X_4 + \beta_{35} X_3 \\
& \times X_5 + \beta_{45} X_4 \times X_5 + \beta_{11} X_1{}^2 + \beta_{22} X_2{}^2 + \beta_{33} X_3{}^2 + \beta_{44} X_4{}^2 + \beta_{55} X_5{}^2
\end{aligned}
\tag{1}
$$

where Y is the predicted xylanase activity (U/mL); $\beta_0$ is the intercept; $\beta_1$, $\beta_2$, $\beta_3$, $\beta_4$, and $\beta_5$ are linear coefficients; $\beta_{12}$, $\beta_{13}$, $\beta_{14}$, $\beta_{15}$, $\beta_{23}$, $\beta_{24}$, $\beta_{25}$, $\beta_{34}$, $\beta_{35}$, and $\beta_{45}$ are interactive coefficients; and $\beta_{11}$, $\beta_{22}$, $\beta_{33}$, $\beta_{44}$, and $\beta_{55}$ are quadratic coefficients.

### 3.4. Enzyme Purification and Identification

For xylanase purification, the enzyme-containing culture supernatant was concentrated with cold ethanol at a ratio of 3:1 (ethanol/supernatant). The sediment was dissolved in Tris-HCl buffer (20 mM, pH 7.2) and then loaded onto a high Q column. A gradient elution of NaCl (0–1 M) was applied to isolate the xylanase. The xylanolytic fractions were dialyzed against Tris-HCl buffer and further purified using a DEAE column, followed by a Sephacryl S-200 column. The molecular weight and purity of the obtained protein were estimated using sodium dodecyl sulfate–polyacrylamide gel electrophoresis (SDS-PAGE) [53]. Liquid chromatography with tandem mass spectrometry (LC-MS/MS) analysis was performed to pinpoint the protein in the band on the SDS-PAGE gel [54].

### 3.5. Enzyme Characterization

The *S. thermocarboxydus* TKU045 xylanase's optimal pH was assessed by measuring xylanolytic activity across a pH range of 4.0 to 10.0, using buffers such as acetate (pH 4.0 and 5.0), phosphate (pH 6.0, 7.0, and 8.0), and Tris-HCl (pH 9.0 and 10.0). To ascertain the ideal temperature, the xylanase assay conducted at varying temperatures from 20 to 80 °C. To explore pH stability, the xylanase was kept at 20 °C for 60 min at various pH values (4.0–10.0) without substrate, and then residual activities were measured at pH 6.0. For assessing thermostability, the leftover activity was determined following individual incubations of *S. thermocarboxydus* TKU045 xylanase at temperatures ranging from 20 to 80 °C for 1 h.

The impact of various chemicals, including Tween 40, Tween 20, Sodium lauryl sulfate (SDS), Triton X-100, (Ethylenedinitrilo)tetraacetic acid (EDTA), $Zn^{2+}$, $Fe^{2+}$, $Fe^{3+}$, $Cu^{2+}$, $Mg^{2+}$, $Mn^{2+}$, $Ba^{2+}$, $Ca^{2+}$, and 2-mercaptoethanol, on xylanolytic activity were explored. Each chemical was introduced singly into the enzyme at a final concentration of 1 mM for 1 h. Subsequently, the effects of the chemicals were evaluated by measuring enzyme activities at pH 6.0 and 70 °C. As control, an enzyme solution without the addition of any chemical was used.

Birchwood xylan, beechwood xylan, oatspelt xylan, cellulose, pectin, and starch were used as the substrates to investigate the substrate specificity of the xylanase.

Various concentrations of xylan (ranging from 0.25 to 3 mg/mL) were utilized to determine the rate of the hydrolysis reaction catalyzed by Xyn_TKU045. Subsequently, a Lineweaver-Burk plot (Figure S4) was employed to calculate the kinetic parameters for xylanase, including $K_m$, $k_{cat}$, and $k_{cat}/K_m$.

Birchwood xylan was used as the substrate to assess the mechanism of hydrolysis of *S. thermocarboxydus* TKU045 xylanase. The hydrolysis products were analyzed at various time intervals (0, 0.5, 1, 2, 3, 4, 5, 6, and 7 h) through high-performance liquid chromatography (HPLC; column: KS-802; solvent: $H_2O$; flow rate: 0.6 mL/min; sample volume: 20 μL; column temperature: 80 °C; refractive index detector). For comparison, a standard mixture containing xylose ($X_1$), xylobiose ($X_2$), xylotriose ($X_3$), xylotetraose ($X_4$), and xylopentose ($X_5$) was utilized.

## 4. Conclusions

This research successfully demonstrated that *S. thermocarboxydus* TKU045 is an efficient xylanase producer using wheat bran-containing medium that could achieve a maximal productivity of 25.314 ± 1.635 U/mL. The isolated xylanase (Xyn_TKU045), characterized by an MW of 43 kDa, was purified and subjected to biochemical analysis from the culture. Xyn_TKU045 displayed optimal activity at pH 6.0 and 60 °C. Identified as an endo-β-1,4-xylanase, Xyn_TKU045 effectively catalyzed xylan into xylotriose and xylobiose as primary products.

**Supplementary Materials:** The following supporting information can be downloaded at: https://www.mdpi.com/article/10.3390/recycling9030050/s1, Figure S1: Two-dimensional (2D) contour plots showing the effect of the amount of wheat bran powder (WBP) and $KNO_3$ (a), WBP and pH (b), WBP and temperature (c), and WBP and incubation time (d). 2D contour plots showing the effect of the amount of $KNO_3$ and pH (e), $KNO_3$ and temperature (f), and $KNO_3$ and incubation time (g). 2D contour plots showing the effect of the pH and temperature (h), pH and incubation time (i), and temperature and incubation time (k); Figure S2: Response surface plots showing the effect of the amount of wheat bran powder (WBP) and amount of $KNO_3$ (a), WBP and pH (b), WBP and temperature (c), and WBP and incubation time (d). Response surface plots showing the effect of the amount of $KNO_3$ and pH (e), and $KNO_3$ and temperature (f), $KNO_3$ and incubation time (g). Response surface plots showing the effect of pH and temperature (h), pH and incubation time (i), and temperature and incubation time (k); Figure S3: Time courses of reducing sugar production generated from the hydrolysis of xylan catalyzed by Xyn_TKU045; Figure S4: Lineweaver-Burk plots of Xyn_TKU045.

**Author Contributions:** Conceptualization and methodology, T.N.T., C.T.D. and S.-L.W.; software, validation, formal analysis, investigation, resources, and data curation, T.N.T., C.T.D., T.D.T., P.H.Q.H., A.D.N., T.K.L.D., T.H.N.D., T.T.U.P. and T.T.L.L.; writing—original draft preparation, T.N.T. and C.T.D.; writing—review and editing, T.N.T., C.T.D. and S.-L.W.; visualization S.-L.W., T.N.T. and C.T.D.; supervision, project administration, and funding acquisition, S.-L.W. and T.N.T. All authors have read and agreed to the published version of the manuscript.

**Funding:** This study was supported in part by a grant from the National Science and Technology Council, Taiwan (NSTC-112-2320-B-032-001), and by a grant from Tay Nguyen University, Vietnam (T2023-43CBTĐ).

**Data Availability Statement:** Data are contained within the article and Supplementary Materials.

**Conflicts of Interest:** The authors declare no conflicts of interest.

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
