# Peer review of "Optimization Production of an Endo-β-1,4-Xylanase from Streptomyces thermocarboxydus Using Wheat Bran as Sole Carbon Source"

_recycling, doi:10.3390/recycling9030050_

Round 1

Reviewer 1 Report

Comments and Suggestions for Authors

1.      The abstract lacks the background of the study. Please mention the background and significance of the study in the abstract.

2.      Briefly introduce the statistical optimization in the introduction section.

3.      What is the novelty of this study. It is a conventional study of enzyme production, purification, and characterization.

4.      Why wheat bran powder and KNO3 were chosen?

5.      The kinetics of the enzyme should also be studied.

6.      Authors are suggested to explain figure 1 and 2 in detail.

7.      Authors are suggested to deduce the 3d structure of the xylanase enzyme using in-silico techniques and carry out Ramachandran analysis for better understanding.

8.      The work is preliminary. Adding the industrial application of the extracted xylanase in the manuscript would add value to the manuscript and strengthen it. 

Author Response

Reviewers' comments:

Reviewer #1:

  1. The abstract lacks the background of the study. Please mention the background and significance of the study in the abstract.

Reply: Thank you for your suggestion. The abstract has been revised. Please see lines 20-22.

  1. Briefly introduce the statistical optimization in the introduction section.

Reply: Thank you for your suggestion. Statistical optimization has been introduced in the introduction section. Please see lines 69-74.

  1. What is the novelty of this study. It is a conventional study of enzyme production, purification, and characterization.

Reply: Please see lines 138-151.

  1. Why wheat bran powder and KNO3 were chosen?

Reply: the selection of wheat bran powder and KNO3 was based on a previous study: [32]. Tran, T.N.; Doan, C.T.; Wang, S.L. Conversion of wheat bran to xylanases and dye adsorbent by Streptomyces thermocarboxydus. Polymers 2021, 13, 287. Please see lines 307-308.

  1. The kinetics of the enzyme should also be studied.

Reply: Thank you for your suggestion. The kinetics of the xylanase have been added. Please see lines 238-246.

  1. Authors are suggested to explain figure 1 and 2 in detail.

Reply: Thank you for your suggestion. Figure 1&2 has been moved to the supporting information section. Since the patterns of these graphs are relatively similar, a general explanation has been made instead of explaining each graph individually. Hope you accept this explanation.

  1. Authors are suggested to deduce the 3d structure of the xylanase enzyme using in-silico techniques and carry out Ramachandran analysis for better understanding.

Reply: We are very grateful for your suggestion. While your suggestion is indeed valuable for enhancing our comprehension of xylanase structure, it's important to note that our study's focus does not entail a comprehensive examination of the xylanase sequence. Therefore, analyzing the xylanase's 3D structure exceeds the scope of our current investigation. Thus, we kindly prefer not to incorporate the suggested content into the manuscript.

  1. The work is preliminary. Adding the industrial application of the extracted xylanase in the manuscript would add value to the manuscript and strengthen it.

Reply: Thank you for your suggestion. The suggested content has been added. Please see lines 270-283.

One more time, we would like to pay our deeply thanks for your careful and valuable comments for enhancement of the quality of the manuscript.

Reviewer 2 Report

Comments and Suggestions for Authors

Manuscript ID: Recycling-2966871

Title: Wheat Bran Valorization to Endo-β-1,4-Xylanase by Streptomyces thermocarboxydus: Production Optimization and Enzyme Purification

The manuscript deals with induction of xylanase in S. hermocarboxydus _TKU045 when grown in the presence of wheat bran and various factors that affect xylanase induction as well as its activity. As such the study is interesting and worthy of publishing. I recommend that authors make some changes so that the results is the paper matches with what is concluded.

1.      The title of paper is problematic – You don’t valorize (convert) wheat bran to xylanase. Rather you grow S. hermocarboxydus _TKU045 in the medium of wheat bran (carbon source) when xylanase production is induced in the organism, in turn the xylanase degrades wheat bran to xylose oligomers.  So the title should appropriately modified to mean what is being done in the paper.

2.      Methods section starting after results and discussion creates a challenge – many a times you are referring to “methods described as above” in the results section, but the results come before the methods section.  Either change sequence of these sections or change descriptions in the results section appropriately.

3.      In several places when you make comparisons, you state data not shown.  In such cases, please include a reference or simply include the values you got in your experiments as a reference point.

4.      Several figures run into multiple pages.  Keping them in a single page with their figure lagend is good for making comparisons.  If necessary, you may break these figures into multiple ones. Alternatively make them smaller so they fit into a single page.

5.      The biggest challenge in this paper is the indiscriminate use of sig-figs.  In the same table you see data with 2 sig-figs all the way to 8 sig-figs.  Sometime zero decimal places going all the way to 4 decimal places.  Please look into your measurements?  Can you make such measurements given the sensitivity of the instruments you are using?  Please correct them appropriately by using correct sig-figs.

Author Response

Reviewers' comments:

Reviewer #2:

The manuscript deals with induction of xylanase in S. hermocarboxydus _TKU045 when grown in the presence of wheat bran and various factors that affect xylanase induction as well as its activity. As such the study is interesting and worthy of publishing. I recommend that authors make some changes so that the results is the paper matches with what is concluded.

  1. The title of paper is problematic – You don’t valorize (convert) wheat bran to xylanase. Rather you grow S. hermocarboxydus _TKU045 in the medium of wheat bran (carbon source) when xylanase production is induced in the organism, in turn the xylanase degrades wheat bran to xylose oligomers. So the title should appropriately modified to mean what is being done in the paper.

Reply: Thank you for your suggestion. The title has been revised.

  1. Methods section starting after results and discussion creates a challenge – many a times you are referring to “methods described as above” in the results section, but the results come before the methods section. Either change sequence of these sections or change descriptions in the results section appropriately.

Reply: Thank you for your suggestion. The manuscript has been checked to correct the errors you pointed out. Please see line 165.

  1. In several places when you make comparisons, you state data not shown. In such cases, please include a reference or simply include the values you got in your experiments as a reference point.

Reply: Thank you for your suggestion. The manuscript has been checked to add the information you suggested. Please see lines 212 and 232.

  1. Several figures run into multiple pages. Keping them in a single page with their figure lagend is good for making comparisons.  If necessary, you may break these figures into multiple ones. Alternatively make them smaller so they fit into a single page.

Reply: Thank you for your suggestion. Figures 1 and 2 have been revised and moved to supporting information section.

  1. The biggest challenge in this paper is the indiscriminate use of sig-figs. In the same table you see data with 2 sig-figs all the way to 8 sig-figs.  Sometime zero decimal places going all the way to 4 decimal places.  Please look into your measurements?  Can you make such measurements given the sensitivity of the instruments you are using?  Please correct them appropriately by using correct sig-figs.

Reply: Thank you for your suggestion. Tables 2 and 3 have been revised following your comment.

One more time, we would like to pay our deeply thanks for your careful and valuable comments for enhancement of the quality of the manuscript.

Round 2

Reviewer 1 Report

Comments and Suggestions for Authors

1.      Check lines 20-21, it should be “byproducts to produce xylanase…”

2.      Lineweaver-Burk plot should be added to the supplementary file.

Author Response

Detailed response to reviewers' comments

 Manuscript ID: recycling-2966871The Original Title: Wheat Bran Valorization to Endo-β-1,4-Xylanase by Streptomyces thermocarboxydus: Production Optimization and Enzyme Purification

Dear Reviewer/Advancer,

We feel pleasure to thank you for your time and effort, as well as your excellent suggestions for refining the readability and impact of the manuscript. We have gone through all the suggestions cautiously and made the revisions accordingly; and all amended parts have been typed in red in the revised manuscript. Finally, we like to express our thanks to your comments and suggestions again. You certainly have served to improve the quality of this paper. We hope our response is satisfactory.

Looking forward to hearing from you.                

Thanking you,

Yours Sincerely,

San-Lang Wang

Director/Professor, Life Sciences Development Center/Department of Chemistry, Tamkang University, Taiwan

Reviewers' comments:

Reviewer #1:

  1. Check lines 20-21, it should be “byproducts to produce xylanase…”

Reply: It has been revised. Please see lines 20-21.

  1. Lineweaver-Burk plot should be added to the supplementary file.

Reply: Thank you for your suggestion. The plot has been added to the supplementary file (Figure S4)

One more time, we would like to pay our deeply thanks for your careful and valuable comments for enhancement of the quality of the manuscript.
